# Noise Aware Finetuning for Analog Non-Linear Dot Product Engine

**Lei Zhao    Luca Buonanno    Aishwarya Natarajan    Jim Ignowski    Giacomo Pedretti**
Hewlett Packard Labs - Artificial Intelligence Research Lab (AIRL)
{lei.zhao, luca.buonanno, aishwarya.natarajan, jim.ignowski,
giacomo.pedretti@hpe.com

## Abstract

As interest in analog acceleration for deep neural networks (DNNs) grows, ReRAM-based Dot-Product Engines (DPEs) offer an energy-efficient solution for performing vector-matrix multiplications (VMMs) in the analog domain. However, DPEs require Analog-to-Digital Converters (ADCs), which contribute significantly to area and power overhead, and rely on digital logic for operations such as non-linear activations. This work presents an ADC-less DNN accelerator that leverages Analog Content Addressable Memory (ACAM) to replace ADCs and digital activation units. By training decision trees to approximate activation functions and programming them to ACAMs, the novel Non-linear DPE (NL-DPE) enables arbitrary activation to be implemented directly in the analog domain, supporting a broader range of future DNN architectures. Additionally, we explore the inherent noise present in real devices of both crossbars and ACAMs and propose noise-aware finetuning techniques that mitigate accuracy loss, demonstrating notable improvements.

## 1   Introduction

Deep neural networks (DNNs) have significantly grown in size and complexity, driving an increasing demand for memory bandwidth and computation. As DNN models continue to expand, the energy consumption associated with frequent data movement between memory and processing units has become a significant bottleneck, known as the *memory wall*. Such bottlenecks have led to the development of In-Memory Computing (IMC) accelerators, where computation occurs directly within the memory. Among the various IMC approaches, ReRAM-based analog computing stands out as one of the most promising solutions due to its potential for high energy efficiency and scalability. ReRAM cells can be organized in a crossbar structure to design Dot-Product Engines (DPEs), which perform vector-matrix multiplications (VMMs) in the analog domain in a single step, achieving low energy consumption and high parallelism [Shafiee et al. (2016); Ankit et al. (2019)].

However, DNNs consist of more than just VMMs; they also require non-linear activations, which are typically executed using digital logic. Therefore, DPE's analog outputs must be converted into digital signals via Analog-to-Digital Converters (ADCs). Unfortunately, ADCs are both energy and area-inefficient, significantly impacting the overall efficiency of ReRAM-based accelerators. ADCs can consume more than 30% of the chip area and account for over 50% of the total power consumption, creating a substantial bottleneck in achieving truly energy-efficient DNN accelerators [Shafiee et al. (2016); Ankit et al. (2019)].

Another challenge for ReRAM-based DNN accelerators is the inherent noise in analog computing. While many existing works tend to overlook or underestimate its effect, our experiments demonstrate that noise plays a crucial role in determining the accuracy of computations on real hardware. In fact, without careful attention to noise, accuracy can degrade to unacceptable levels, as noted also in other

38th Second Workshop on Machine Learning with New Compute Paradigms at NeurIPS 2024(MLNCP 2024).

studies [Mao et al. (2022)]. Although some efforts have been made to address this issue [Momeni et al. (2024)], novel computation primitives may require to carefully handle new sources of noise.

In this work, we propose the Non-linear DPE (NL-DPE), an analog DNN accelerator design that eliminates the need for ADCs and digital activation units. We convert activation functions into decision trees (DTs), which are then mapped onto the programmable Analog Content Addressable Memory (ACAM) for analog computation. Since ACAM accepts analog inputs and produces digital outputs, ADCs are no longer required after the crossbars. To address the reliability issues of analog computation, we measure noise directly from real devices and develop detailed noise models that account for various sources, including programming inaccuracies and conductance fluctuations, etc. Noise models are integrated into a finetuning process to minimize the accuracy gap caused by noise.

## 2 Analog Computing with ReRAM

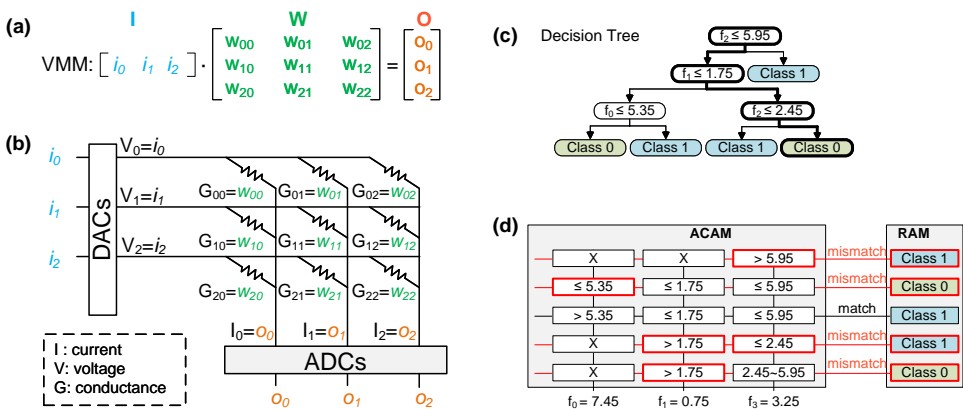

Figure 1: **(a)** An example of VMM computing. **(b)** Computing the VMM in DPE. **(c)** An example of a DT. **(d)** Mapping the DT onto ACAM and performing an inference.

Fig. 1(a) shows an example VMM operation ($\mathbf{O} = \mathbf{I} \cdot \mathbf{W}$). $\mathbf{O} \in \mathbb{R}^3$, $\mathbf{I} \in \mathbb{R}^3$ and $\mathbf{W} \in \mathbb{R}^{3\times3}$. Fig. 1(b) illustrates computing the VMM operation in the DPE with a 3×3 crossbar. A ReRAM cell is placed at every intersection of the horizontal wires and vertical wires. Weight elements ($W_{ij}$) are programmed as ReRAM conductance ($G_{ij}$), and input elements($i_i$) are represented by voltage ($V_i$) on horizontal wires. Following Kirchhoff's law and Ohm's law, the current ($I$) from vertical wires convey the dot-product result of the input vector and the weight matrix. The analog VMM operation is completed in a single step within the DPE, achieving high computing parallelism.

Recently, ReRAM has been used to build ACAMs, accelerating tree-based machine learning algorithms in the analog domain [Li et al. (2020); Pedretti et al. (2021a, 2023)]. Unlike digital CAM, which is limited to comparing a single input bit with a stored bit, an ACAM cell can compare an analog input value against a stored analog range. To map a trained DT of Fig. 1(c) onto an ACAM array, we traverse each leaf node back to the root, storing the feature thresholds along the path in a row of ACAM cells (Fig. 1(d)). For example, the last row in the ACAM array stores the feature thresholds corresponding to the highlighted path in the DT. Wildcard cells ('X') indicate that the particular feature is irrelevant along that path, thus the full range is programmed. Given an input feature vector during inference, if all feature values fall within the ranges stored in a row, the corresponding match line will be activated, retrieving the predicted class from an adjacent RAM. ACAMs have thus analog input, but digital output representing a match of mismatch.

## 3 Non-Linear Dot Product Engine

Fig. 2 shows a two-layer snippet from a larger neural network mapped to a conventional DPE (a) and the proposed NL-DPE (b). The first layer is followed by a Sigmoid activation function, while the second layer has no activation. Digital-to-Analog Converters (DACs) are required to convert the digital input into analog signals for computing VMM with the weights in the crossbar. In conventional

DPE-based accelerators (Fig. 2(a)), activations are typically performed in the digital domain, thus ADCs are then needed to convert the analog output from the crossbars back into digital form. In our proposed NL-DPE (Fig. 2(b)), we use ACAM to compute activations. As ACAM accepts analog input, ADCs are no longer needed after the crossbars. The output of our ACAM is a digital signal, DACs are still needed before the next DPE. To replace the ADC, ACAM is programmed as an identity function, a special case of activation. For instance, in the figure, the second layer has no activation, so we use an ACAM implementing the identity function to replace the second ADC.

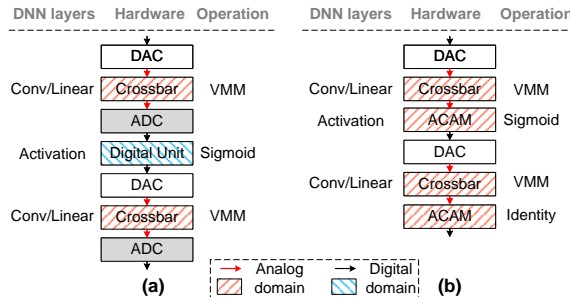

Fig. 3(a) demonstrates how ACAM is used to compute the Sigmoid activation with a 3-bit output. In this example, we assume 0 and 1 are quantized to $000_b$ and $111_b$ respectively, and the remaining 6 values are evenly distributed between 0 and 1, as shown by the y axis in Fig. 3(a). Note that any arbitrary output bit format can be used; the incremental progression from $000_b$ to $111_b$ is simply for illustration purposes. Each output bit ($y_2$, $y_1$, and $y_0$) is computed using a separate DT, treating each as a binary classification task. The training dataset contains only one feature

Figure 2: Mapping DNN on **(a)** converntional DPE and **(b)** NL-DPE.

(input $x$), with the output bit as the target. For example, to train the DT for the second most significant bit (MSB) ($y_1$), the target is set to 1 when the input $x$ falls within the range of -0.91 to 0.28, or when $x$ exceeds 1.79, as shown in Fig. 3(a). Fig. 3(b) depicts the trained DT and Fig. 3(c) its mapping onto an ACAM array. Unlike conventional DTs, which generalize to predict unseen inputs, the DTs here memorize the exact patterns from the training data, forcing overfitting. Note that the output of the ACAM already represents the bit value, there is no need to access to an attached memory as opposed to a Look Up Table (LUT) approach [Zhu et al. (2022)].

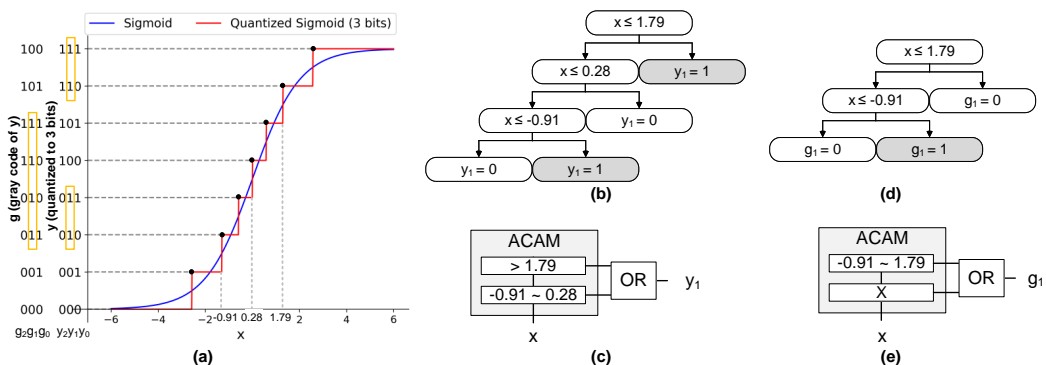

Figure 3: **(a)** Analytical and 3-bit quantized sigmoid function using conventional binary ($y$) and Grey code ($g$)format.**(b)** Trained DT to predict the second MSB of the activation with conventional binary representation ($y_1$) and **(c)** its mapping on ACAM. **(d)** Trained DT to predict the second MSB of the activation with Grey coding ($g_1$) and **(e)** its mapping on ACAM.

Since activations are typically monotonic functions, the less significant bits in the output tend to toggle more frequently between 0 and 1, leading to deeper DTs. To mitigate this, we propose an encoding scheme based on Gray code to reduce bit toggling. Fig. 3(a) also shows the Gray code for the output $y$ on the left (denoted as $g$), where only one bit changes between consecutive values. Fig. 3(d) shows the DT trained using Gray code as the output format and Fig. 3(e) its mapping to ACAM. Theoretically, all DTs except for the MSB can be reduced to half their original size, leading to 50% resource usage reduction.

To enable subsequent computations, the Gray code output from the ACAM must be converted back to its original binary form. The conversion only needs simple XOR gates, as illustrated in the Appendix Section A.

## 4 Noise Aware Finetuning for ACAM

ReRAM stores information via its conductance but suffers from significant analog non-idealities, including noise during programming and reading phases. We developed a detailed noise model, calibrated with real-device measurements, to capture these effects. Additionally, given ACAM's more complex cell structure compared to the simple 1T1R crossbar cell, we also propose a noise model for ACAM's threshold values. The detailed noise models are described in Appendix B.

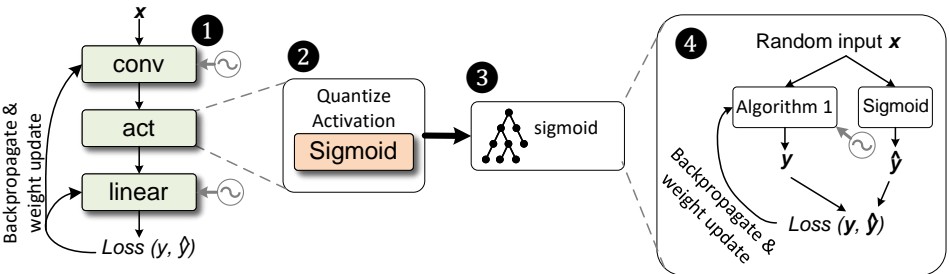

Figure 4: Steps of converting a pretrained DNN into a noise-resilient model that will be deployed on NL-DPE.

---

**Algorithm 1:** Differentiable Approximation of ACAM

**Input:** $x$: one input value for finetuning; $\mathbf{w}_i^L$: lower thresholds of the DT for $i$th output bit; $\mathbf{w}_i^H$: higher thresholds of the DT for $i$th output bit;

**Output:** $y$: an 8-bit output of the ACAM

**Data:** $g_{ratio}$: threshold-to-conductance ratio; $g_{min}$: minimum conductance of RRAM; $g_{max}$: maximum conductance of RRAM; $\epsilon$: a very small positive constant to avoid division-by-zero.

1 **for** $i = 0$ **to** 7 **do**
2    $\mathbf{g}_i^L = \text{Clip}(\mathbf{w}_i^L \cdot g_{ratio} + g_{min}, [g_{min}, g_{max}])$;
3    $\mathbf{g}_i^H = \text{Clip}(\mathbf{w}_i^H \cdot g_{ratio} + g_{min}, [g_{min}, g_{max}])$;
4    $\tilde{\mathbf{g}}_i^L = \text{Noise}(\mathbf{g}_i^L)$; $\tilde{\mathbf{g}}_i^H = \text{Noise}(\mathbf{g}_i^H)$;
5    $\tilde{\mathbf{w}}_i^L = (\tilde{\mathbf{g}}_i^L - g_{min})/g_{ratio}$;
6    $\tilde{\mathbf{w}}_i^H = (\tilde{\mathbf{g}}_i^H - g_{min})/g_{ratio}$;
7    $\mathbf{m}_i = \text{ReLU}(x - \tilde{\mathbf{w}}_i^L) \cdot \text{ReLU}(\tilde{\mathbf{w}}_i^H - x)$;
8    $m_i = \text{Sum}(\mathbf{m}_i)$;
9    $m_i = m_i/(m_i + \epsilon)$;
10 **end**
11 $y = 0$;
12 **for** $i = 7$ **to** 0 **do**
13    **if** $i == 7$ **then**
14      $b_i = m_i$;
15    **else**
16      $b_i = (m_i - b_{i+1})^2$;
17    **end**
18    $y = y + b_i \cdot 2^i$;
19 **end**

---

To address noise in crossbars, we adopt the recently proposed *Analog Slicing* (AS) [Pedretti et al. (2021b); Song et al. (2024)] method to map *unquantized* DNN weights onto crossbars. Compared to the traditional *Bit Slicing* (BS) [Shafiee et al. (2016)] approach, AS reduces the noise impact on crossbars to negligible levels. A detailed analysis of BS and AS can be found in Appendix C.

The remaining of this section will focus on the Noise Aware Finetuning (NAF) to mitigate the noise impact on ACAMs.

Fig. 4 illustrates the overall process of converting a pretrained DNN into a noise-resilient model. Given a pretrained DNN model, step ❶ performs a small number of additional training iterations (typically fewer than 10) to mitigate the noise in the crossbar. During each iteration, each Convolutional and Linear layer's weight matrix is converted into a MSC matrix and LSC matrix (see Appendix C for details), and their corresponding error matrices are sampled based on the noise model described in Appendix B. Then, we use Eq. 9 to get the noise-injected weight matrix. This step finetunes the model to be resilient against noise inherent in crossbar operations.

Because activation functions will be implemented in ACAM, which produces fixed point output, we need to quantize the output of activation functions in step ❷. The quantization will generate two parameters for each activation, i.e., the scaling factor and zero point, which will be used in step ❸ to generate the training data and labels for training DTs for each output bit of the activation functions. We use scikit-learn to train the DTs.

Step ❹ addresses noises in ACAM by performing NAF for each DT independently. Unlike step ❶, this finetuning performs individual DTs rather than the entire DNN. Taking the sigmoid operation as an example, random input $x$ is used to compute its ground truth output $\hat{y}$. The same input $x$ is processed through Algorithm 1, which performs a differential approximation of the ACAM computation Pedretti et al. (2022). In Algorithm 1, the DT thresholds are organized into trainable tensors, which can be updated using gradient-based optimization methods such as stochastic gradient descent (SGD). When computing Algorithm 1 in the forward pass, a random noise sampled from the noise model is injected into the DT thresholds. Finally, $\hat{y}$ and $y$ are used to compute a loss, which is used in the backward pass and update the DT thresholds.

Algorithm 1 demonstrates the differentiable method for computing an 8-bit output activation using ACAMs. In this setup, 8 DTs compute 1 bit each of the final output. We organize the lower and upper thresholds of each DT into tensors and treat these tensors as trainable parameters. To enable gradient computation, all operations on these tensors are made differentiable.

The for loop in lines 1–10 calculates each of the 8 output bits while incorporating noise into the thresholds. Since the noise model described in Appendix B is based on conductance, the loop first converts the DT thresholds into their corresponding target conductance values based on a threshold-to-conductance ration ($g_{ration}$) (lines 2–3). Noise sampled from the model is then added to the target conductance (lines 4). The noisy conductance values are subsequently converted back into DT thresholds (lines 5–6). In line 7, we replace the comparison operation between inputs and thresholds with a ReLU function. This ensures that $\mathbf{m}_i$ is positive if the input $x$ lies within the range defined by the lower and upper thresholds; otherwise, $\mathbf{m}_i$ is zero. Line 8 uses a Sum() operation as a differentiable replacement for the OR gate in ACAM, ensuring $m_i$ is positive if any value in $\mathbf{m}_i$ is positive. Line 9 employs a division to quantize $m_i$ to 0 or 1, with a small $\epsilon$ added to prevent division by zero. After the first loop, each $m_i$ is a floating-point number very close to either 0 or 1, representing the Gray code output bit of an ACAM array.

The second loop, in lines 12–19, implements a differentiable version of the XOR-based decode logic. Here, we replace the XOR operation with a combination of subtraction and squaring to maintain differentiability.

As a result, all computations in Algorithm 1 are fully differentiable, allowing gradients of the threshold tensors to be computed during backpropagation.

## 5 Results

Table 1 shows the accuracy of various models at different stages of NAF. The balled numbers corresponds to the steps depicted in Fig. 4. A small error is introduced by the crossbars even in the presence of noise (step ❶), thanks to the mitigation effect of AS. Using DT-based activations (step ❸) doesn't introduce significant error as well, making the approach potentially generalizable to other accelerators that may efficiently perform inference of tree-based models **?**. However, adding noise to the ACAM performing DT inference (step ❹) has an enormous impact on the model accuracy due to the non-linear operation of the ACAM that exacerbates the noise effect, which can only be recovered thanks to NAF. Thus, results demonstrate that our DT-trained activations approach is feasible and does not impact accuracy, but NAF is needed for practical utilization with analog accelerators

Table 1: Accuracy of various stages in the proposed NAF.

| Model | LeNet | ResNet-18 | SENet | EfficientNet | ResNet-34 | VGG11 | ShuffleNet-v2 | DenseNet121 |
|---|---|---|---|---|---|---|---|---|
| Dataset | MNIST | Cifar10 | Cifar10 | Cifar10 | Cifar10 | ImageNet | ImageNet | ImageNet |
| Baseline (FP32) | 99.02 | 92.59 | 95.4 | 91.17 | 73.302 | 70.38 | 69.356 | 74.438 |
| ❶ | 99.04 | 92.52 | 95.4 | 91.14 | 73.174 | 70.218 | 68.902 | 74.154 |
| ❶ + NAF | 99.04 | 92.52 | 95.4 | 91.17 | 73.3 | 70.22 | 69.01 | 74.154 |
| ❷ | 99.03 | 92.48 | 94.67 | 90.81 | 73.154 | 69.962 | 68.542 | 72.394 |
| ❸ | 99.02 | 92.32 | 93.86 | 89.52 | 72.922 | 69.356 | 68.22 | 72.42 |
| ❹ | 10.36 | 71.12 | 61.92 | 54.06 | 41.944 | 50.53 | 61.696 | 53.231 |
| ❹ + NAF | 99.01 | 91.8 | 93.23 | 89.05 | 72.066 | 67.4 | 68.05 | 71.34 |

NL-DPE completely removes the ADC and digital activation unit in conventional DPE accelerators. We compare the power and area between our proposed NL-DPE computing unit and conventional DPE-based computing unit (including ADC and activation units needed by DPE) in Table 2. We take

the ADC data from [Shafiee et al. (2016)]. For digital activation unit, we take the data from FlexSFU [Reggiani et al. (2023)], a piecewise linear (PWL) based implementation to approximate arbitrary activation in digital domain. Because ACAM array produces the output bits directly, without the need of associated RAM (see Fig. 1(d)) as opposed to other ACAM-based approaches [Zhu et al. (2022)], ACAM's area is significantly smaller than its digital counterpart FlexSFU. As FlexSFU only accepts digital inputs, which requires power- and area-expensive ADCs, NL-DPE's area and power are both lower than that of DPE considering the whole unit.

Table 2: Power and area breakdown of DPE and NL-DPE.

| | **Params** | **Spec** | **Power** ($mW$) | **Area** ($\mu m^2$) |
|---|---|---|---|---|
| DPE computing unit | | | | |
| Crossbar | array size
number | $256 \times 256$
4 | 0.82 | 4325 |
| ADC Shafiee et al. (2016) | resolution
frequency
number | 8bits
1.2GSps
256 | 512 | 307200 |
| FlexSFU Reggiani et al. (2023) | bit width | 8bits | 6.6082 | 26431 |
| Total | | | 519.42 | 337956 |
| NL-DPE computing unit (This work) | | | | |
| Crossbar | array size
number | $256 \times 256$
4 | 0.82 | 4325 |
| ACAM | array size
number | $130 \times 1$
256 | 40.95 | 6973 |
| Total | | | 41.77 | 11301 |

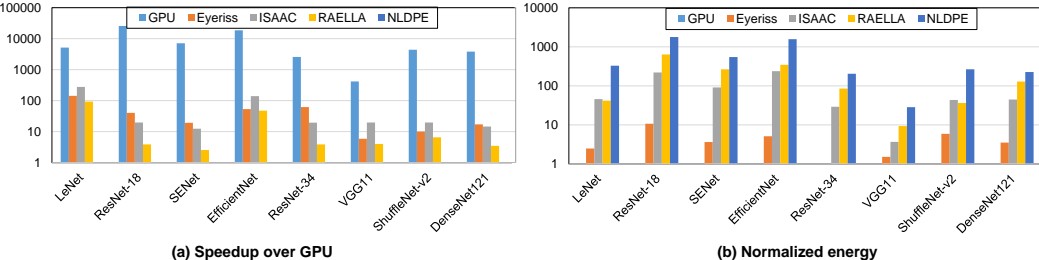

(a) Speedup over GPU            (b) Normalized energy

Figure 5: Speedup and normalized energy of running the benchmarks on different accelerators.

Fig. 5 compares running the benchmarks on our proposed NL-DPE with two state-of-the-art ReRAM-based accelerators (ISAAC [Shafiee et al. (2016)] and RAELLA [Andrulis et al. (2023)]) as well as a NVIDIA H100 GPU. ISAAC and RAELLA use conventional DPEs for VMMs. Because ISAAC and RAELLA only supports simple activation functions (e.g., ReLU), we extend them with FlexSFU for more complex activations (e.g., SiLU, tanh, etc.). Notably, NL-DPE outperforms GPU in energy efficiency by a geometric mean factor of 4187 while being $292 \times$ faster. At the same time, NL-DPE beats ISAAC and RAELLA thanks to the improved energy efficiency of NL-DPE compared to conventional bulky ADCs and to digital activation unit for the case of all models tested. The latency improvement of NL-DPE compared to ISAAC and RAELLA is due to a drastic reduction of the digital operation needed in the post-processing of dot products.

## 6 Conclusion

We presented the NL-DPE, a novel ADC-less analog in-memory computing primitive. NL-DPE builds on previous works on crossbar arrays and ACAM, by connecting them in the analog domain, with the former accelerating dot products and the latter activation functions, effectively implementing $f(xW)$ in the analog domain. Activations are approximated with DTs which the ACAM can efficiently accelerate. We propose a Noise Aware Finetuning (NAF) routing to increase accuracy in the presence of ReRAM analog noise and we benchmark the proposed accelerator against the conventional DPE and GPU, reaching significant improvements. We envision NL-DPE as a new building block for in-memory computing, opening up new possibilities for model design fully exploiting the programmable analog non-linearity operation.

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

# A    Reconstruction of Gray coded output

Conversion from Gray code to binary can be achieved using simple logic gates: each binary bit is obtained by performing an XOR operation with all the higher-order bits in the Gray code representation, except the MSB which is the same between these two formats, i.e.,

$$b_i = \begin{cases} g_i & i = n-1 \\ XOR(g_{n-1}, g_{n-2}, ..., g_{i+1}) & i < n-1 \end{cases}$$

where $b_i$ and $g_i$ are the $i$th bit in the binary and Gray code format. $n$ is the output bit width. The overhead of these XOR gates are included in the power and area estimation of ACAM.

# B    ReRAM Noise Model

The ReRAM noises come from two main sources: (1) a ReRAM cell may not be precisely programmed to the desired conductance levels, due to the stochastic nature of the ionic migration during the programming operation; (2) each read operation from a programmed ReRAM cell may yield slightly varying conductance values, due to various physical phenomena, such as thermal and random telegraph noise. In addition, ReRAM demonstrates minimal conductance drift after programming, which is neglected in this paper.

It is challenging to build an accurate noise model for ReRAM-based full analog accelerators. First, both programming and fluctuation noises are conductance-dependent and thus difficult to model. For example, the conductance fluctuation for high resistance states shows a constant relative standard deviation, which decreases linearly once the cell transitions to a low resistance state after forming a strong filament Ielmini (2016). Second, the crossbars and ACAM Units have different cell structures, i.e., while DPE adopts 1T1R crossbar structure, each ACAM cell consists of multiple ReRAM cells and transistors. The cell noise behaviors are thus also very different.

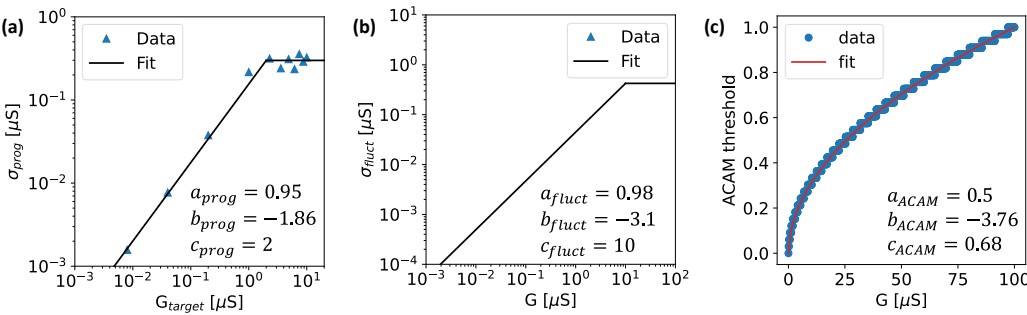

Figure 6: Experimental data and model fitting of **(a)** standard deviation of programmed conductance $\sigma_{prog}$ as a function of the target conductance $G_{target}$, **(b)** standard deviation of fluctuated conductance $\sigma_{fluct}$ as a function of the mean conductance $G$, and **(c)** transfer function of the ACAM, namely ACAM threshold as a function of the programmed conductance $G$.

In this paper, we develop the model by fitting it to noise data collected from real ReRAM devices Sheng et al. (2019). We run a program-and-verify algorithm on Ta-Ox ReRAM devices using a tolerance of $\pm 0.55\mu S$ for $G_{target} > 1\mu S$ and tolerance proportional to the conductance levels for $G_{target} \leq 1\mu S$. We then read the conductance values 1000 times to assess the fluctuation noise.

Fig. 6(a) indicates that the maximum standard deviation of programming noise $\sigma_{prog}$ is approximately $0.4\mu S$ and can be modeled as a function of the target conductance $G_{target}$. Note that in practice, since an inference accelerator programs the weights of its trained model once, we may further minimize programming noise by using a smaller programming tolerance at the cost of a longer programming time. Fig. 6(b) indicates that the standard deviation of read fluctuation $\sigma_{fluct}$ can be modeled as a function of the mean conductance $G$. The experimental results match those reported in Mao et al. (2022), where the log-scale standard deviation increases approximately linearly with the mean until it saturates at a certain point, which may be related to the threshold between low and high conductance states.

We model the programming and read noises using normal distribution and compute their standard deviation as follows.

$$\sigma_x = \exp(a_x \log(G.\text{clip}(0, c_x) + b_x) \tag{1}$$

We fit the parameters $a_x$, $b_x$ and $c_x$ to the experimental data and report them in Fig. 6(a,b). The programming and read fluctuation errors are then computed as $G_{write} = \sigma_{prog} \cdot \mathcal{N}(0, 1)$ and $G_{read} = \sigma_{fluct} \cdot \mathcal{N}(0, 1)$, respectively, where $\mathcal{N}(0, 1)$ is a normal distribution with mean 0 and standard deviation of 1.

Given a ReRAM cell that was programmed to its target conductance $\mathbf{G}_{target}$, we compute the readout conductance as follows:

$$\mathbf{G} = \mathbf{G}_{target} + \mathbf{G}_{write} + \mathbf{G}_{read}. \tag{2}$$

The conductance model in Eq. 2 applies to ReRAM cells in both crossbars and ACAMs. However, ACAM involves a non-linear operation so the relation between conductance and the ACAM threshold needs to be modeled as well. Fig. 6(c) reports the experimental data on the ACAM threshold values as a function of the programmed conductance Pedretti et al. (2021a). We then model the transfer function as

$$\text{TH} = \exp(a_{ACAM} * \log(G) + b_{ACAM}) + c_{ACAM} \tag{3}$$

with the fitting parameters reported in Fig. 6(c).

## C   Noise Analysis for Bit Slicing and Analog Slicing

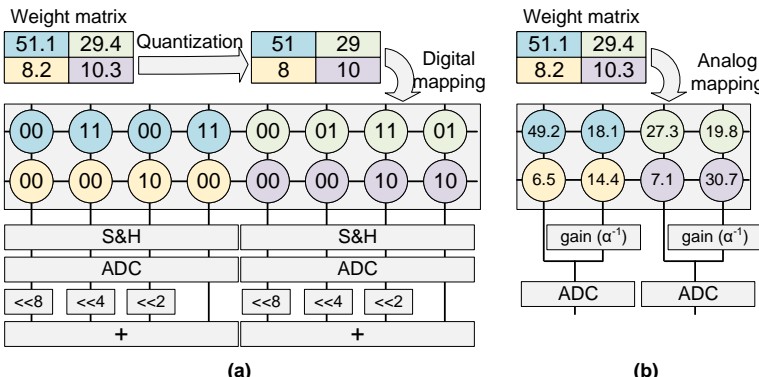

Figure 7: Mapping a weight matrix onto crossbars with **(a)** Bit Slicing and **(b)** Analog Slicing.

Typically, Bit Slicing (BS) involves quantizing the full-precision floating-point DNN weights into fixed-point format (e.g., 8 bits) and using ReRAM cells to store portions of the quantized bits. Taking the first value of the weight matrix in Fig. 7(a) as an example, $w = 51.1$ is first quantized into an 8-bit fixed point value (i.e., $w_q = 51 = 00110011_b$, for simplicity, we assume a simple nearest-integer quantization for illustration purpose). The quantization error is $\epsilon_q = w_q - w = 0.1$. Each ReRAM cell stores 2 bits, so four cells are needed to store each weight. The first cell stores the most two significant bits ($B_3 = 00$), the next cell stores the next two bits ($B_2 = 11$), and so on ($B_1 = 00$, $B_0 = 11$). Therefore, the quantized weight value 51 is represented by:

$$w_q = B_3 \cdot 64 + B_2 \cdot 16 + B_1 \cdot 4 + B_0 \tag{4}$$

Now, assuming there is an error on each cell $\epsilon_i$, the actual stored value in these four cells are actually:

$$\tilde{w}_q = (B_3 + \epsilon_3) \cdot 64 + (B_2 + \epsilon_2) \cdot 16 + (B_1 + \epsilon_1) \cdot 4 + B_0 + \epsilon_0 \tag{5}$$

Therefore, the total error of BS is:

$$\epsilon_{BS} = \tilde{w}_q - w = \tilde{w}_q - w_q + \epsilon_q = \epsilon_3 \cdot 64 + \epsilon_2 \cdot 16 + \epsilon_1 \cdot 4 + \epsilon_0 + \epsilon_q \tag{6}$$

The error in each cell is amplified by the significance of the stored bits. For instance, the error in $B_3$ is amplified by $64\times$.

Fig. 7(b) shows the same weight matrix mapped to the crossbar with Analog Slicing (AS). We still take the first weight value as an example. First, a most significant cell (MSC) is programmed targeting the conductance $51.1\mu S$. Because of the programming and read errors introduced in Appendix B, assuming the actual conductance that we could read out from MSC is $g_{MSC} = 49.2\mu S$, resulting in an error $\epsilon_{MSC} = 51.1 - 49.2 = 2.7\mu S$.

After programming the MSC, the resulting error is mapped into a second ReRAM cell, namely the least significant cell (LSC). In order to exploit the full conductance range, $\epsilon_{MSC}$ is scaled up by multiplying a gain $\alpha$, which is determined by:

$$\alpha = \frac{g_{max}}{\max \epsilon_{MSC}} \tag{7}$$

where the denominator is the maximum error of all MSCs in the matrix, $g_{max}$ is maximum conductance that can be programmed in ReRAM cells. Thus, the LSC corrects the $\epsilon_{MSC}$ error by being programmed to a conductance that represents the scaled-up error. In this example, assuming $\alpha = 10$, so the LSC is programmed targeting $27\mu S$. Again, there may be also an error in LSC, the actual conductance of the LSC might be $g_{LSC} = 18.1\mu S$, the error is $\epsilon_{LSC} = 27 - 18.1 = 8.9\mu S$. During inference, both cells receive the input at the same time, and their output currents are combined, with the LSC's current scaled down to compensate for the error with a gain circuit, attenuating it by $\alpha^{-1}$. The actual stored value in MSC and LSC considering the errors is:

$$\begin{aligned} \tilde{w} &= g_{MSC} - \alpha^{-1} \cdot g_{LSC} \\ &= (w + \epsilon_{MSC}) - \alpha^{-1} \cdot (\alpha \cdot \epsilon_{MSC} + \epsilon_{LSC}) \\ &= w - \alpha^{-1} \cdot \epsilon_{LSC} \end{aligned} \tag{8}$$

Thus, the final error for AS is

$$\epsilon_{AS} = \tilde{w} - w = \alpha^{-1} \cdot \epsilon_{LSC} \tag{9}$$

In this example, the final noise is thus $51.1 - (49.2 + 18.1 \div 10) = 0.09\mu S$.

Because $\epsilon_{MSC}$ is always much smaller than $g_{max}$, $\alpha$ is always larger than 1. Therefore, comparing Equation 6 and 9, the error in analog slicing is significantly smaller.

In summary, AS has three advantages over BS: Firstly, the weights no longer need quantization, which introduces quantization error. Secondly, as shown in the previously analysis, AS has less final error than BS. Lastly, AS needs less ReRAM cells than BS.

