# OpenReview forum: "Noise Aware Finetuning for Analog Non-Linear Dot Product Engine"
_NeurIPS.cc/2024/Workshop/MLNCP — MLNCP Poster_

### Official Review · Reviewer_1Tuo · 2024-09-19
**Interesting solution the ADC problem in analog accelerators**

**Rating:** 7
**Confidence:** 4

**Review:**

The paper proposes to get rid of the Analog to Digital conversion needed after an analog vector matrix multiplication done with a Resistive memory crossbar and use instead an analog content addressable memory to implement the non linear function applied after the matmul in deep neural networks.

The paper is clearly written and has both experimental results on a fabricated crossbar as well as simulations quantifying the performance of such a DNN when all the components are added together.

I have a few comments that could improve the paper:
- Is it not a bit misleading to say that the ADC is replaced when the ACAM is essentially a more general component that can also function as an ADC (Line 79)? It seems from reading the paper that the ACAM is also a way to do the ADC, but apparently more efficiently according to Table 2, and it can also implement the non linearity.
- It is not super clear when reading the paper what is purely simulation and what is actually measured on the crossbar. For instance how is Figure 6 derived? Is it extrapolation from other measurements to what training would cost or is it actually running those models with crossbars?
- Why is there an ADC in Fig4a, isn't the circuit before already digital?
- Typos: "approch", "converntional"

---

### Official Review · Reviewer_EobY · 2024-09-21
**Analog activation functions are very interesting, yet the noise aware finetuning procedure should better not be used in some cases**

**Rating:** 6
**Confidence:** 3

**Review:**

**Summary**. While analog crossbars, a.k.a "Dot Product Engines" (DPEs, which employ ReRAM devices to store the weights), are known to be extremely efficient to perform vector-matrix multiplication (VMMs), so that mapping a pre-trained model onto an analog chip may result in significant power and area gains. However, the advantages brought by analogue devices can largely be offset by the need to convert from analog to digital and back to analog when passing the DPE output through activation functions *when mapped into digital*. This paper proposes a solution with the following contributions:

i) instead of mapping the activation function into digital with the usual $DAC \to DPE \to ADC \to activation \to DAC$ flow, the authors propose to approximate the activation function as a decision tree and map the resulting tree onto an *analog content addressable memory* (ACAM). The resulting implementation eliminates the need for ADCs at the output of the DPEs, such that it reads as $DAC \to DPE \to ACAM \to DAC$ -- DACs are still needed as the output of ACAMs are digital. The system is called a *Nonlinear DPE* (NL-DPE). Details about how layers are mapped onto NL-DPEs, namely analog slicing and how the binary decision trees are designed, are provided inside Sections 3 and 4.

ii) Taking into account the *analog noise* inherent to the ReRAM devices as well as that of the ACAMs, the authors propose a *Noise-Aware" fine-tuning (NAF) procedure to mitigate the loss of accuracy due to this noise -- it is the analog counterpart of quantization aware training (QAT). The procedure is detailed inside Section 5.

iii) The proposed NL-DPE, as well as the associated NAF strategy are tested on three architectures (LeNet, ResNet-18 and ResNet-50) on two datasets (MNIST and CIFAR-10).

**Strengths**.

- The paper is clearly written and easy to follow.
- It is great to have such hardware-oriented papers at a conference like NeurIPS. I learned, or re-learned, things about analog hardware I had forgotten about and it's great to communicate to the broader ML audience problems inherent to the use of analog hardware.
- The proposed implementation of analog activation functions is interesting and seems promising.

**Weaknesses**.

- The most important weakness in my eyes is that we can observe from Table 1 that the proposed NAF procedure **worsens** the performance of ResNets on CIFAR-10 when introducing ACAM noise, and only marginally improves the performance when the system is only subject to the DPE's noise.
- Why not evaluating ResNets on the ImageNet dataset?
- I'm not sure to understand why decision trees need to be learned given that the analytical form of the activation functions is already known. Can't we find a systematic way to hardcore any given activation function into a decision tree? If not, why is this difficult? I think it's worth highlighting.

**Overall appreciation**.
I find the core idea of implementing activation functions in analog very interesting. However, with the title raising high expectations on the "noise aware fine-tuning" procedure, it is quite disappointing to see that it significantly worsens the model performance, meaning that it is better not to apply this procedure at all. In spite of these flaws, I do bear in mind it is work in progress and the paper is still very much worth sharing!:)

---

### Decision · Program_Chairs · 2024-10-10

Accept (Poster)